# Prospective Assessment of Tumour Burden and Bone Disease in Plasma Cell Dyscrasias Using DW-MRI and Exploratory Bone Biomarkers

**DOI:** 10.3390/cancers15010095

**Published:** 2022-12-23

**Authors:** Gaurav Agarwal, Guido Nador, Sherin Varghese, Hiwot Getu, Charlotte Palmer, Edmund Watson, Claudio Pereira, Germana Sallemi, Karen Partington, Neel Patel, Rajkumar Soundarajan, Rebecca Mills, Richard Brouwer, Marina Maritati, Aarti Shah, Delia Peppercorn, Udo Oppermann, Claire M. Edwards, Christopher T. Rodgers, Muhammad Kassim Javaid, Sarah Gooding, Karthik Ramasamy

**Affiliations:** 1Department of Clinical Haematology, Oxford University Hospitals NHS Foundation Trust, Oxford OX3 7LE, UK; 2Oxford Translational Myeloma Centre, Oxford OX3 7LD, UK; 3Botnar Research Centre, The Nuffield Department of Orthopaedics Rheumatology and Musculoskeletal Sciences, University of Oxford, Oxford OX3 7LD, UK; 4Department of Radiology, Oxford University Hospitals NHS Foundation Trust, Oxford OX3 9DU, UK; 5Oxford Centre for Magnetic Resonance, Radcliffe Department of Medicine, University of Oxford, Oxford OX3 9DU, UK; 6Department of Radiology, Hampshire Hospitals NHS Foundation Trust, Hampshire SO22 5DG, UK; 7Nuffield Department of Surgical Sciences (NDS), Oxford OX3 9DU, UK; 8Department of Clinical Neurosciences, University of Cambridge, Cambridge CB2 0QQ, UK; 9MRC Molecular Haematology Unit, Weatherall Institute of Molecular Medicine, University of Oxford, Oxford OX3 9DS, UK

**Keywords:** multiple myeloma, MGUS, biomarkers, bone disease, DW-MRI

## Abstract

**Simple Summary:**

Whilst multiple myeloma (MM) remains incurable, two clinical priorities are to prolong remission and reduce complications, of which fragility fractures are a major source of morbidity. To this end, there is the need to develop biomarkers that can accurately track tumour burden and bone loss to guide treatment decisions. Here, we conducted a pilot feasibility study exploring the value of novel serum bone turnover and plasma cell burden markers and Diffusion-Weighted Magnetic Resonance Imaging (DW-MRI) when added to standard clinical assessment in patients with MM, monoclonal gammopathy of undetermined significance (MGUS) and smouldering MM (SMM). We show serum DKK1 and BCMA as possible correlates of tumour burden, and that serum sclerostin may correlate with bone mineral density. Furthermore, we validate DW-MRI in longitudinal assessment of tumour volume. Our study highlights emerging serum and radiological biomarkers for assessment of tumour burden and bone loss, which require further study in larger cohorts to validate these findings and understand their clinical utility.

**Abstract:**

Novel biomarkers for tumour burden and bone disease are required to guide clinical management of plasma cell dyscrasias. Recently, bone turnover markers (BTMs) and Diffusion-Weighted Magnetic Resonance Imaging (DW-MRI) have been explored, although their role in the prospective assessment of multiple myeloma (MM) and monoclonal gammopathy of undetermined significance (MGUS) is unclear. Here, we conducted a pilot observational cohort feasibility study combining serum BTMs and DW-MRI in addition to standard clinical assessment. Fifty-five patients were recruited (14 MGUS, 15 smouldering MM, 14 new MM and 12 relapsed MM) and had DW-MRI and serum biomarkers (P1NP, CTX-1, ALP, DKK1, sclerostin, RANKL:OPG and BCMA) measured at baseline and 6-month follow-up. Serum sclerostin positively correlated with bone mineral density (*r* = 0.40−0.54). At baseline, serum BCMA correlated with serum paraprotein (*r* = 0.42) and serum DKK1 correlated with serum free light chains (*r* = 0.67); the longitudinal change in both biomarkers differed between International Myeloma Working Group (IMWG)-defined responders and non-responders. Myeloma Response Assessment and Diagnosis System (MY-RADS) scoring of serial DW-MRI correlated with conventional IMWG response criteria for measuring longitudinal changes in tumour burden. Overall, our pilot study suggests candidate radiological and serum biomarkers of tumour burden and bone loss in MM/MGUS, which warrant further exploration in larger cohorts to validate the findings and to better understand their clinical utility.

## 1. Introduction

Multiple myeloma (MM) is the second most common haematological malignancy, characterised by the malignant proliferation of plasma cells [1]. Despite an increasing range of treatment options that have increased overall survival, patients eventually relapse and require multiple rounds of therapy [2]. Given the long clinical course and follow-up, there is a need for reliable biomarkers, in addition to secreted monoclonal proteins, that can track disease burden and bone disease, in order to inform the timing and intensity of treatment. Such markers may additionally capture a subset of smouldering MM (SMM) patients that may benefit from earlier treatment [3,4] and refine monitoring frequency in monoclonal gammopathy of undetermined significance (MGUS) [5]. Currently, serum paraprotein and free light chains are widely used for disease monitoring; however, utility is limited in non-paraprotein-secreting MM, and although they track the secretory capacity of a plasma cell clone, that feature can change over the course of disease. To this end, novel serum and imaging biomarkers have been explored, although their clinical value in the prospective assessment of an extended spectrum of plasma cell dyscrasias remains unclear.

In recent years, preclinical studies have shown pathological bone turnover marker (BTM) signalling between MM and bone stromal cells as a driver of MM bone disease [6]. Namely, osteocytes secrete sclerostin [7] and MM cells (as well as bone and stromal cells) release Dickkopf-1 secreted glycoprotein (DKK1) [8], both are Wnt inhibitors that suppress osteoblast-mediated bone formation. Furthermore, MM cells induce increased receptor activators of nuclear factor kappa B ligand (RANK-L) expression and inhibit production of its antagonist osteoprotegerin (OPG) by stromal cells, favouring osteoclast-mediated bone destruction [9,10]. Given the central role of MM cells in paracrine modulation of local BTM levels, and that BTMs in peripheral blood correlate with bone marrow levels for some of these [11], it is feasible that serum BTM measurement may reflect global MM tumour burden. Furthermore, these biological insights may provide novel disease-specific measures of MM bone loss, in order to target patients with established MM, as well as those at risk of acquiring bone lesions for bone-sparing therapy. However, the use of BTMs has been largely restricted to the experimental setting, and so their clinical utility in the prospective assessment of MGUS and MM remains unclear.

Alongside serum biomarkers, whole-body Diffusion-Weighted Magnetic Resonance Imaging (DW-MRI) is gaining traction as a radiological measure of disease burden. Compared with morphological MRI sequences, contrast is generated based on the diffusion of water molecules, generating superior visual discrimination between healthy and diseased bone marrow, and enabling focal lesions to be quantified for their apparent diffusion coefficient (ADC) [12]. The quantitative capabilities and superior sensitivities [13] have led the International Myeloma Working Group (IMWG) to propose DW-MRI for use in general clinical practice [12,14]. Recently, the Myeloma Response Assessment and Diagnosis System (MY-RADS) guidelines were published [15] with standardised criteria for scoring MM disease burden and response assessment. However, there is limited published data for DW-MRI in the context of prospective staging in new and relapsed MM, and its clinical value in conjunction with serum BTMs remain unexplored.

We conducted a single-centre observational cohort feasibility study to pilot the combined value of novel BTMs and DW-MRI in the prospective assessment of MM tumour burden and bone disease. We recruited a cohort of patients covering the clinical spectrum of plasma cell dyscrasias, including MGUS, SMM, newly diagnosed MM and relapsed MM. Participants had DW-MRI and a panel of serum biomarkers measured at baseline diagnosis and 6-month follow-up, correlated with standard clinical assessment metrics of tumour burden and bone loss. Whilst overall sample numbers were low, our pilot work suggests the utility of DW-MRI imaging in the longitudinal assessment of tumour burden and, additionally, suggests candidate BTMs in the prospective assessment of MM disease activity and associated MM bone disease that warrant further exploration.

## 2. Materials and Methods

### 2.1. Study Recruitment

We conducted a single-centre observational prospective cohort study (ClinicalTrials.gov Identifier: NCT03951220). Intended to be a powered feasibility pilot study, 67 participants were recruited between March 2018–March 2020 across: (i) healthy volunteers (*n* = 12); (ii) MGUS (*n* = 14, recruited between 18–6511 days following date of diagnosis); (iii) smouldering MM (*n* = 15, recruited between 31–2219 days following date of diagnosis); (iv) newly diagnosed multiple myeloma (NDMM) (*n* = 14, recruited between 6–154 days following date of diagnosis); and (v) relapsed MM (*n* = 12, recruited 0–245 days following clinical definition of relapse).

Participants were eligible to be included in the trial if they met all of the following criteria: able to and willing to give informed consent; male or female, aged 18 years or above; patient attending Oxford NHS Haematology–Oncology centre; NDMM or newly relapsed MM eligible for next therapy, or SMM or intermediate–high-risk MGUS (with clinical diagnostic criteria set by IMWG). The following exclusion criteria were applied: unable or unwilling to give informed consent; women who may be pregnant, breast feeding or women of child-bearing potential unwilling or unable to take sufficient precautionary measures were excluded due to Dual-energy X-ray absorptiometry (DXA) imaging; signs of spinal cord compression; patients with documented metastatic lesions from another type of malignancy; and known contraindication for MRI scan, including unacceptable pain on lying flat for 1 h.

### 2.2. Study Assessment

Following recruitment, all participants attended a baseline study appointment, and only MGUS/MM patients had follow-up at six months. At each visit, all participants had a DW-MRI scan, and MGUS/MM patients additionally had a DXA scan and had peripheral blood drawn for analysis of serum biomarkers and standard myeloma bloods. Peripheral blood was collected as a morning fasted 20 mL clotted (serum) sample and stored at −20 °C prior to analysis. As participants were recruited at either diagnosis or relapse, any standard investigations their clinician deemed necessary for diagnostic evaluation within NHS protocols were also performed.

Unfortunately, due to clinical urgency, most patients with new or relapsed myeloma had commenced chemotherapy imminently after their diagnosis before they were able to attend baseline assessment. Previous studies have suggested that bone turnover markers and DW-MRI ADC of focal lesions can change with therapy, therefore, to standardise intercurrent chemotherapy as a potentially confounding factor, we excluded the minority of patients not on chemotherapy at point of baseline assessment from analysis.

### 2.3. Clinical Correlates

Electronic health records were used to derive patient demographics and disease characteristics (including haemoglobin, serum creatinine, serum calcium and history of osteoporosis). Percentage plasma cells were measured from the diagnostic bone marrow specimens. Patients were classified by International Myeloma Working Group (IMWG) response criteria as a correlate of therapy response [16]. DXA scans were reported for BMD and T-scores at femoral neck (FN) and the mean values across lumbar vertebrae 1–4 (L1-4).

### 2.4. Serum Biomarker Analysis

Established serum biomarkers in routine clinical use were analysed by standard National Health Service (NHS) laboratory protocols: P1NP (Type I Procollagen N-terminal Peptide), CTX-1 (Type I Collagen Cross-Linked C-Telopeptide) and ALP (Alkaline Phosphatase). Exploratory serum biomarkers were analysed by commercial enzyme-linked immunosorbent assay (ELISA) in triplicate, with outliers excluded and mean values taken: SOST (sclerostin), DKK1 (Dickkopf WNT Signalling Pathway Inhibitor 1), ratio of RANKL (receptor activator of nuclear factor kappa-B ligand) to OPG (osteoprotegerin) and BCMA (B-cell maturation antigen).

### 2.5. DW-MRI Analysis

DW-MRI was conducted on the Siemens 3 Tesla Prisma (software version VE11C) for all patients. MRI scans were double reported by expert radiologists in line with the 2018 MY-RADS guidelines, as detailed by Messiou and colleagues [15]. In brief, the pattern of MM disease was assessed qualitatively; in patients with a focal lesion, the ADC was quantified at baseline and follow-up scans. For response assessment, baseline and follow-up DW-MRI scans were compared and semi-quantitative criteria were used to assign a Response Assessment Category (RAC): 1 (highly likely to be responding), 2 (likely to be responding), 3 (no change), 4 (likely to be progressing) and 5 (highly likely to be progressing).

### 2.6. Statistical Analysis

Visualisation of data distribution, as well as the Shapiro–Wilk and Kolmogorov–Smirnov tests were used to test data sets for normal distributions. Inter-group differences between two sets of unpaired and paired data were analysed using unpaired and paired *t*-tests, respectively. Inter-group differences for three or more sets of data were analysed using the Kruskal–Wallis test or Analysis of Variance (ANOVA) tests for non-parametric and parametric data, respectively, with post hoc tests if appropriate. Spearman’s rank correlation coefficient was used to assess correlations between continuous variables. Fisher–Freeman–Halton exact test was used to compare categorical variables in an RxC contingency table, where sample sizes were small. Longitudinal % change in bone biomarkers, DXA BMD and novel MR measurements were calculated as a ratio (follow-up measurement divided by paired baseline measurement for the same participant).

## 3. Results

### 3.1. Study Design and Participant Demographics

Table 1 shows the baseline patient and disease characteristics (measures of myeloma disease as well as bone health) for 55 patients who were enrolled. The median age of all participants was 66.2 (range 40.9—85.3) years, and there was no difference in mean ages (Kruskal–Wallis, *p* = 0.24) or male to female ratios between groups (Fisher–Freeman–Halton exact test, *p* = 0.37).

Of 14 patients with new myeloma, 8 were transplant eligible and 6 were transplant ineligible. Of 12 patients with relapsed myeloma, all patients had indolent first relapses following first-line therapy. Of 15 patients with SMM, 8 had low-risk, 6 had intermediate-risk and 1 had high-risk SMM, as defined by the 2020 IMWG criteria [17]. At baseline (or after but before follow-up visit), 13/14 new myeloma and 9/12 relapsed myeloma patients had received active treatment.

In terms of disease characteristics, there was no difference between groups in the ISS stage (Fisher–Freeman–Halton exact test, *p* = 0.53) or baseline haemoglobin (ANOVA, *p* = 0.14), serum creatinine (Kruskal–Wallis, *p* = 0.44) or serum calcium (Kruskal–Wallis, *p* = 0.99). There was a statistically significant difference between paraprotein immunoglobulin type between groups (Fisher–Freeman–Halton exact test, *p* = 0.01), with greater proportion of IgA-type paraproteins in new myeloma and MGUS and greater light chain only secretory type in smouldering myeloma. There was a quantitative difference in mean paraprotein immunofixation levels between patients with relapsed myeloma (8.9 g/L) and MGUS (9.0 g/L) compared with new myeloma (18.1 g/L) and smouldering myeloma (19.1 g/L); however, these differences did not reach statistical significance (Kruskal–Wallis, *p* = 0.09).

In terms of bone changes, there was no difference in baseline DXA T-scores between groups for either lumbar spine (ANOVA, *p* = 0.33) or femoral neck (Kruskal–Wallis, *p* = 0.73) between groups. There was no difference in the percentage of patients with radiological evidence of previous fractures (Fisher–Freeman–Halton exact test, *p* = 0.29), or percentage of patients with known osteoporosis (Fisher–Freeman–Halton exact test, *p* = 0.41) between groups. Of 14 patients with MGUS, 4 had radiological evidence of a previous fracture (3 vertebral and 1 non-vertebral), although none of these patients later had lytic lesions identified in DW-MRI. As expected, there was a statistically significant difference in the number of patients on bone protection agents at baseline between groups (Fisher–Freeman–Halton exact test, *p* < 0.001), with a greater proportion of new or relapsed myeloma patients on active or previous therapy.

### 3.2. Correlation between Bone Turnover Markers and Bone Mineral Density in MGUS/MM

A panel of seven serum biomarkers were measured in patients with MGUS, SMM, new and relapsed MM: serum BCMA and a set of six BTMs selected based on their known role as paracrine mediators between MM and bone microenvironment cells (Figure 1A). Firstly, the differences in serum biomarkers were analysed between disease groups (Appendix A). With the exception of P1NP (greater in MGUS than relapsed MM (*p* < 0.01)) and SOST (greater in SMM than new MM (*p* < 0.05)), there were no differences in baseline serum biomarkers between the spectrum of plasma cell dyscrasia populations.

Next, correlations between individual BTMs were analysed in a pooled cohort of MGUS and MM patients (Figure 1B). BTMs were classed as known markers of bone formation (P1NP and ALP) and bone resorption (CTX-1, SOST, DKK1 and higher RANK:OPG ratio), and Spearman’s rank correlations were calculated between individual BTM pairs. There was moderate positive correlation between P1NP with CTX-1 (*r* = 0.70, *p* < 0.001) and ALP (*r* = 0.37, *p* < 0.01). Serum sclerostin had weak positive correlation with CTX-1 (*r* = 0.32, *p* < 0.05) and weak negative correlation with the RANKL:OPG ratio (*r* = −0.29, *p* < 0.05).

We explored the relationship between BTMs and bone mineral density (BMD) measured by DXA in a pooled cohort of MGUS and MM patients. Serum sclerostin had moderate positive correlation with BMD at lumbar spine (*r* = 0.54, *p* < 0.001) (Figure 1C) and femoral neck (*r* = 0.40, *p* < 0.01) (Figure 1D), suggesting that greater BMD is associated with greater serum sclerostin. Furthermore, serum sclerostin was lower in patients with previous radiological evidence of fracture (127 pg/mL vs. 176 pg/mL, Mann–Whitney, *p* < 0.05) (Figure 1E) and previous history of osteoporosis (118 pg/mL vs. 153 pg/mL, Mann–Whitney, *p* < 0.05) (Figure 1F).

### 3.3. Plasma Cell Burden Biomarkers Can Track Tumour Burden in MM

We examined whether serum biomarkers could measure tumour burden at baseline assessment. There was moderate positive correlation between serum paraprotein with P1NP (Pearson’s *r* = 0.49, *p* = 0.01) and BCMA (Pearson’s *r* = 0.42, *p* = 0.03) in a pooled cohort of MGUS/MM patients (Figure 2A). Furthermore, in patients with light chain SMM/MM, combined serum free light chains (cSFLC) were positively correlated with serum DKK1 [Pearson’s *r* = 0.67, *p* < 0.05] and ALP [Pearson’s *r* = 0.78, *p* < 0.01] (Figure 2B). There was no difference in serum biomarker measurements between patients with standard-risk and high-risk cytogenetics [defined as ≥1 of: t(4;14), t(14;16), t(14;20), gain/amp(1q21) or del(17p)] [18], in any myeloma (new, relapsed or smouldering) [Mann–Whitney, *p* > 0.05]. Collectively, these results suggest that candidate serum biomarker measurements correlate with known plasma cell burden markers, suggesting they may reflect tumour volume.

Next, we considered whether serum biomarkers could track tumour burden serially (Figure 2C). The longitudinal % change in serum biomarkers was compared between responders (Partial Response or better) and non-responders, as per IMWG criteria based on serial change in serum paraprotein. Analysis was performed in a pooled cohort of patients with SMM/MM with paired measurements between baseline and follow-up (in which active treatment was received by 10/11 new myeloma, 7/9 relapsed myeloma and 0/12 SMM patients, with the remaining under observation). The longitudinal change differed between IMWG-defined responders and non-responders for serum DKK1 (37% decrease vs. 11% increase, *p* < 0.05) and serum BCMA (83% decrease vs. 29% decrease, *p* < 0.01). These results indicate that serial changes in DKK1 and BCMA may capture longitudinal changes in tumour burden.

### 3.4. MY-RADS Can Assess Baseline and Longitudinal MM Tumour Burden from DW-MRI

We explored whether MY-RADS analysis of DW-MRI could categorise changes in tumour burden. As discussed under ‘Methods’, due to clinical urgency, most patients had commenced chemotherapy imminently after diagnosis and before baseline DW-MRI. To minimize variability due to this factor, we included only patients who were on chemotherapy at time of baseline DW-MRI. Of patients with new or relapsed myeloma with lytic lesions identified on DW-MRI, it had been between 6–81 days since initiating chemotherapy.

In patients with new and relapsed MM, those with at least one focal lesion identified on DW-MRI had higher % plasma cells in their diagnostic bone marrow aspirate compared with those with no lesions (median: 8% vs. 18%, Mann–Whitney, *p* = 0.02) (Figure 3A). Next, we tested whether the diffusivity of focal lesions (ADC) correlated with serum markers of tumour burden. In our dataset, there was no correlation between ADC and serum paraprotein or any BTM (Pearson’s, *p* > 0.05). However, DW-MRI lesion ADC had moderate negative correlation with serum BCMA, both at baseline (Pearson’s *r* = −0.55, *p* = 0.03) and with % change longitudinally (Pearson’s *r* = 0.87, *p* = 0.02) (Figure 3B). Collectively, these findings suggest that focal bone lesions with lower ADC on DW-MRI are associated with greater serum BCMA, a known marker of increased tumour burden.

Next, we explored MY-RADS Response Assessment Criteria (RAC) as a tool to longitudinally measure changes in tumour burden. The longitudinal % change in serum biomarkers was compared between MY-RADS responders (RAC 1 or 2) and non-responders (RAC ≥ 3). The longitudinal change differed between MY-RADS responders and non-responders for serum DKK1 (55% decrease vs. 16% increase, *p* < 0.05) (Figure 3C). Finally, we tested concordance between MY-RADS RAC and IMWG response criteria (Figure 3D). The observed numbers of patients for each IMWG Response Group differed between MY-RADS RAC groups (Fisher–Freeman–Halton exact test, *p* = 0.015). The greatest percentage of IMWG responders were classified as MY-RADS 1 (highly likely to be responding), whilst the greatest percentage of IMWG non-responders were classified as MY-RADS 3 (stable), suggesting that MY-RADS RAC differed between therapy responders and non-responders, in agreement with IMWG Response Groups.

## 4. Discussion

In this study, we conducted a single-centre prospective observational cohort study to evaluate the value of DW-MRI and serum BTMs in prospectively assessing tumour burden and associated bone disease when added to standard clinical assessment in an extended spectrum of plasma cell dyscrasia. This work was intended as a pilot feasibility study, hence there were relatively few patients recruited, and definitive conclusions cannot be drawn. Nonetheless, our results suggest three key observations that now warrant validation in larger cohorts.

Firstly, our data highlight serum sclerostin as a positive correlate of BMD in MGUS and MM patients. Sclerostin negatively regulates bone formation through Wnt antagonism [19]; consistent with this, antibody-based sclerostin inhibitors increase bone mass in mouse models of MM [20], and therefore a negative correlation may be expected. Nevertheless, a positive correlation between sclerostin and BMD has additionally been noted in chronic kidney disease [21,22,23], postmenopausal osteoporosis [24,25] and rheumatoid arthritis [26]. It has been speculated that sclerostin may be an adaptive response to higher bone formation [27]. Furthermore, given that sclerostin is largely osteocyte-specific [20], and that its levels in bone marrow and peripheral blood are correlated [11], higher serum sclerostin may reflect greater total bone volume or osteocyte pool. Additional work is required to validate this association and to understand whether sclerostin may be used to screen an eligible population for bone-sparing therapy.

Secondly, we validated the potential utility of bone turnover and plasma cell burden serum biomarkers for assessment of MGUS/MM tumour burden. We found that the BTMs P1NP and ALP correlated with paraprotein, although the clinical value of these markers may be limited due to their non-specific nature. As more targeted markers, our data validate that serum BCMA and DKK1 correlate with serum paraprotein and cSFLC, respectively, at baseline assessment. This is consistent with DKK1 as a secreted protein and BCMA as a cell surface receptor of plasma cells [28,29]. Moreover, serial measurements over 6-month follow-up could distinguish between IMWG-defined responders and progressors, in line with previous reports [30,31,32,33]. Given that myeloma is currently incurable, with many patients having regular blood monitoring, it is important to have sensitive and specific markers. Whilst serum paraprotein is most commonly used for this purpose, this fails in non- or oligo-secretory MM, and additionally, the secretory capacity of a plasma cell clone can change during the disease. Whilst our findings now need to be replicated in larger cohorts, our data suggest adjunct markers such as DKK1 and BCMA may be used in conjunction to track tumour burden, depth of response and early relapse.

Thirdly, we validated MY-RADS recommendations of DW-MRI reporting [15] for assessment of MM tumour burden. We found that patients with focal lesions had greater BM% plasma cells and that lower ADC measurements were associated with higher serum BCMA (both at baseline and with longitudinal change), consistent with higher plasma cell burden. This is in line with ADC being inversely proportional to cellular density in other cancers [34,35] and lower ADC values associated with tumour volume in both MM [36] and metastatic bone lesions from solid tumours [37]. Furthermore, we found that MY-RADS RAC can detect longitudinal changes in tumour load, in agreement with the IMWG classification of therapy response, validating previous work [38]. Given the lack of objective measures of MM disease burden, MY-RADS RAC could therefore provide serial radiological assessment in adjunct to serum paraprotein for monitoring therapy response after diagnosis. However, the prognostic value of MY-RADS remains unclear [39,40], and future work should now assess whether MY-RADS RAC scoring can track disease severity over longer-term periods and aim to understand its clinical utility.

Our study has a number of strengths. Firstly, participants were analysed prospectively and recruited from a range of myeloma subtypes, as well as MGUS, enabling us to assess a range of outcomes simultaneously in a wide spectrum of plasma cell dyscrasias. Secondly, patient assessment was performed in parallel with NHS care, enabling us to assess the additional burden and value in a real-world setting.

Our study has some notable limitations. Firstly, this work was intended to be a feasibility pilot study; loss to follow-up was further compounded by a vulnerable patient population shielding over the COVID-19 pandemic, therefore leading to a long recruitment period. Small population sizes may explain discrepancies between the literature and some reported results; for example, our MGUS cohort did not have statistically greater rates of fracture compared with MM, and ADC did not correlate with serum paraprotein. Secondly, due to difficulty in timing the baseline investigations around new diagnoses which warranted urgent chemotherapy, most patients at baseline had already commenced chemotherapy. As a result, it was not possible to obtain bone turnover marker and DW-MRI data pretherapy, which may have blunted the effect sizes when analysing longitudinal changes between baseline and follow-up in these metrics. Thirdly, there were limitations in using serum paraprotein as the disease burden gold standard correlate, given that this fails in non-secretory myeloma, and the secretory capacity of plasma cells can vary.

## 5. Conclusions

In summary, we report results from a pilot feasibility study testing the value of joint BTMs and DW-MRI in prospective assessment of plasma cell dyscrasias. Whilst sample numbers were low, our data suggest several observations. The study highlights plasma cell burden markers DKK1 and BCMA and validates DW-MRI MY-RADS RAC as biomarkers that can potentially track tumour burden at baseline and longitudinally. Additionally, we find serum sclerostin as a possible correlate of BMD in a cohort of MGUS/MM. Overall, our study highlights radiological and serum biomarkers of tumour burden and bone loss in MM/MGUS that warrant further exploration in larger-scale studies to validate these findings and better understand their clinical utility.

## Figures and Tables

**Figure 1 cancers-15-00095-f001:**
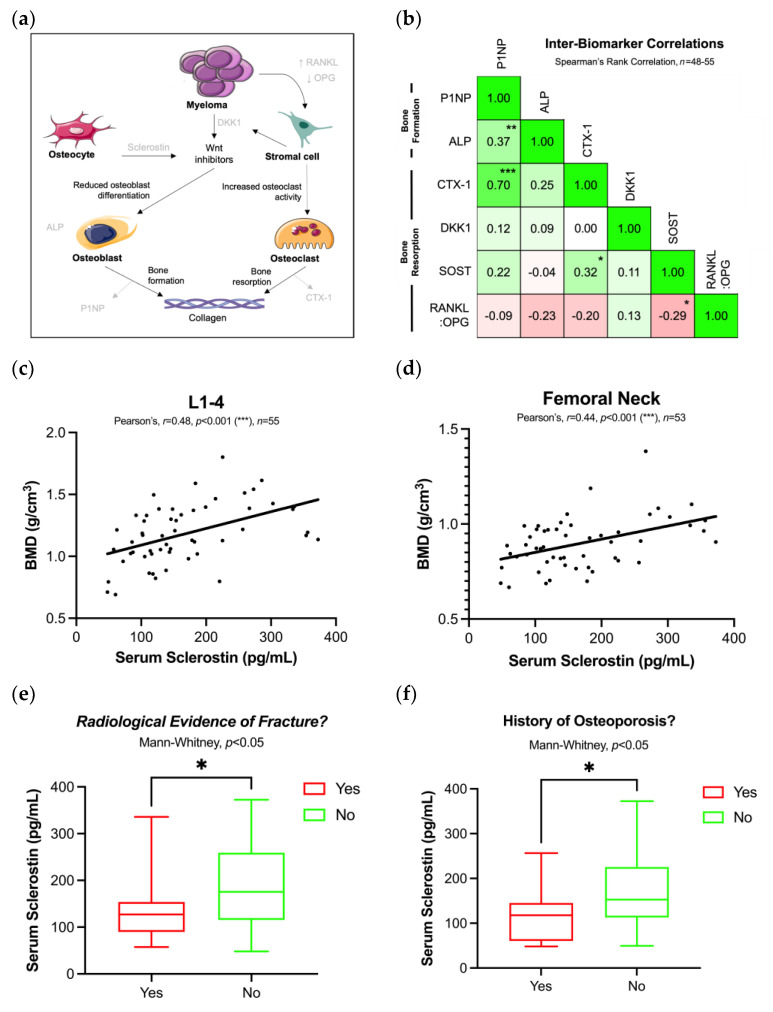
Correlation between bone turnover markers and bone mineral density in MGUS/MM. (**a**) Basic schematic of bone turnover markers investigated in context of paracrine signalling between MM and bone microenvironment cells (interactions not comprehensive). (**b**) Correlation matrix between baseline measurements of serum bone turnover markers in a pooled cohort of MGUS and MM patients. The figures represent Spearman’s rank correlation coefficients between biomarkers; statistically significant correlations are marked. (**c**,**d**) Serum sclerostin positively correlates with bone mineral density at lumbar spine (L1-4) and femoral neck in a pooled cohort of MGUS/MM patients at baseline visit. (**e**) Serum sclerostin is lower in MGUS/MM patients with radiological evidence of previous fracture at baseline assessment. (**f**) Serum sclerostin is lower in MGUS/MM patients with history of previous osteoporosis, assessed at baseline assessment. * *p* < 0.05, ** *p* < 0.01, *** *p* < 0.001.

**Figure 2 cancers-15-00095-f002:**
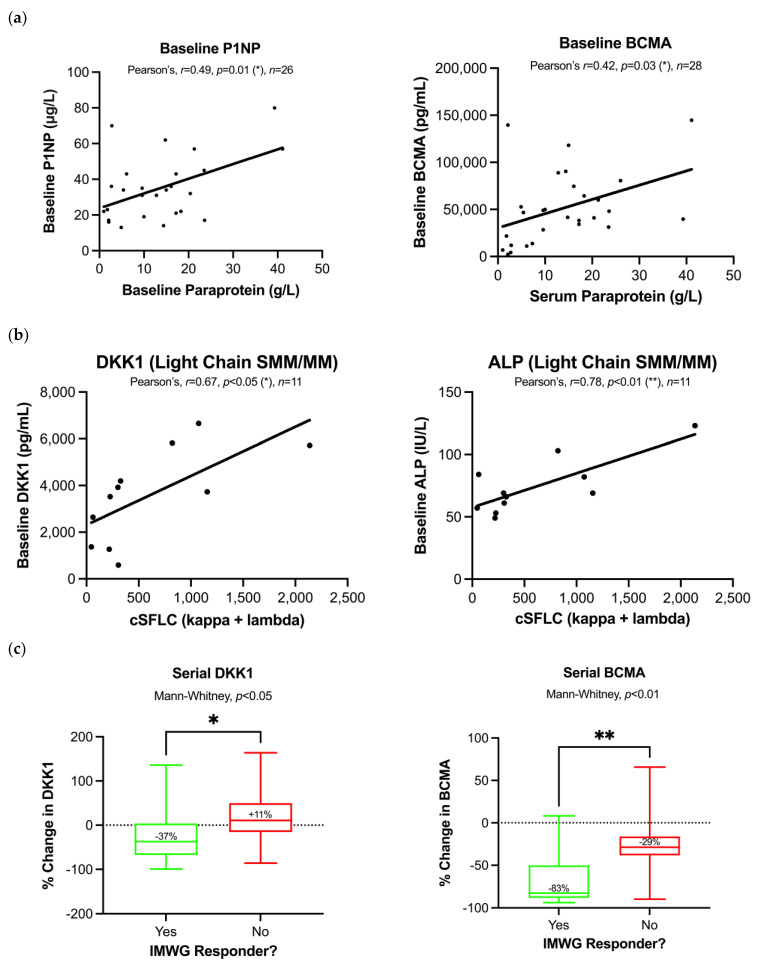
Serum bone turnover and plasma cell burden biomarkers track tumour burden in MM. (**a**) Serum biomarkers were measured and correlated with serum paraprotein at baseline assessment. There was statistically significant positive correlation between P1NP/BCMA and paraprotein, suggesting that these biomarkers can track tumour burden and bone loss. (**b**) In light chain SMM/MM, the combined sum of lambda and kappa serum free light chains (cSFLCs) has a positive correlation with serum DKK1 and ALP measurements. (**c**) The longitudinal % change in serum biomarkers was compared between responders (Partial Response or better) and non-responders, as per IMWG criteria based on serial change in serum paraprotein, in all patients with SMM/MM (*n* = 30−32) under observation or therapy. * *p* < 0.05, ** *p* < 0.01.

**Figure 3 cancers-15-00095-f003:**
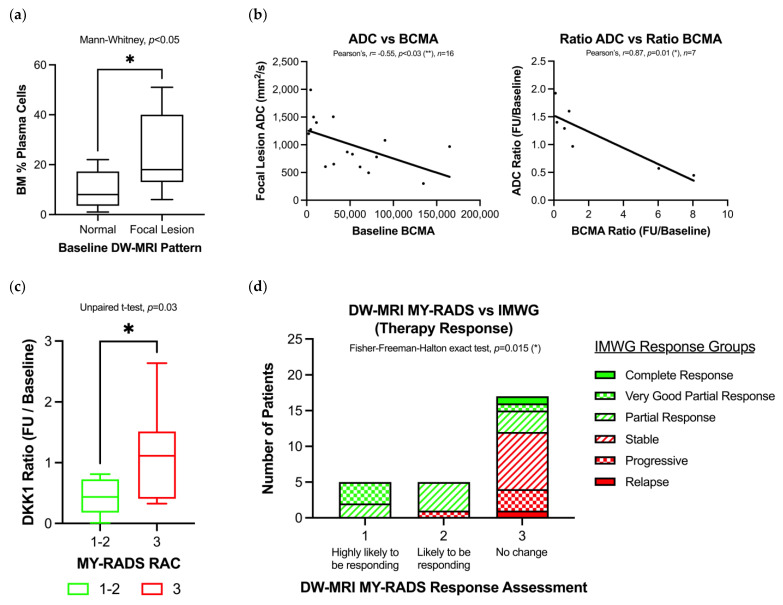
MY-RADS RAC from DW-MRI can assess longitudinal changes in MM tumour burden. (**a**) Bone marrow plasma cell % compared between patients with focal lesion or normal appearance on baseline DW-MRI scan. (**b**) Negative correlation between focal lesion ADC measurements and serum BCMA at baseline and longitudinally (ratio expressed as follow-up/baseline measurements). (**c**) Longitudinal change in DKK1 compared between MY-RADS responders (RAC 1–2) and non-responders (RAC 3). (**d**) From serial DW-MRI analysis, MY-RADS RAC of therapy response correlates in agreement with IMWG Response Group (Fisher–Freeman–Halton exact test, *p* = 0.015). The greatest percentage of IMWG responders were classified as MY-RADS 1 (highly likely to be responding), whilst the greatest percentage of IMWG non-responders were classified as MY-RADS 3 (stable). * *p* < 0.05, ** *p* < 0.01.

**Table 1 cancers-15-00095-t001:** Baseline patient and disease characteristics.

Variable	Subgroup	New MM	Relapsed MM	SMM	MGUS	*p*-Value
**Patient Demographics**						
**Total**		**14**	**12**	**15**	**14**	
**Sex**	Female (%)	2 (14)	4 (33)	5 (33)	7 (50)	0.37
	Male (%)	12 (86)	8 (67)	10 (67)	7 (50)	
**Age (years (median range))**		63.5 (40.9–80.7)	70.8 (56.8–85.3)	65.6 (54.3–82.5)	65.5 (52.1–82.3)	0.78
**Disease characteristics**						
**ISS Stage**	I	7 (50)	6 (50)	7 (46)	-	0.53
	II	3 (21)	3 (25)	4 (27)	-	
	III	3 (21)	2 (17)	0 (0)	-	
	N/A	1 (8)	1 (8)	4 (27)	-	
**Haemoglobin (g/L)**		119.6	119.3	130.6	128.3	0.14
**Serum creatinine (μmol/L)**		81.8	90.5	80.5	92.3	0.44
**Serum calcium (mmol/L)**		2.40	2.37	2.39	2.39	0.99
**Paraprotein type**	IgG	8 (57)	9 (75)	10 (67)	8 (57)	0.01 **
	IgA	5 (36)	0 (0)	0 (0)	4 (29)	
	IgM	0 (0)	0 (0)	0 (0)	2 (14)	
	Light chain	1 (7)	3 (25)	5 (33)	0 (0)	
**Mean Paraprotein** **(g/L [Range])**		18.1 (4.9–39.3)	8.9 (1.0–26.4)	19.1(10–41.6)	9.0 (2.0–21.3)	0.09
**Bone characteristics**						
**Known osteoporosis (%)**		2 (14)	2 (17)	4 (27)	3 (21)	0.41
**Radiologic evidence of previous # (%)**	None	7 (50)	5 (42)	12 (80)	10 (71)	0.29
	Vertebral	5 (36)	5 (42)	3 (20)	3 (22)	
	Non-vertebral	1 (7)	0 (0)	0 (0)	1 (7)	
	Mixed	1 (7)	2 (16)	0 (0)	0	
**Baseline DXA T-score**	L1-4	−0.76	0.63	−0.12	−0.41	0.33
	Femoral neck	−1.09	−1.18	−1.03	−1.44	0.73
**On bone protection**	Current	7 (50)	6 (50)	3 (20)	2 (14)	<0.001 ***
	Previous	0 0)	5 (42)	0 (0)	0 (0)	
	Never	7 (50)	1 (8)	12 (80)	12 (86)	

Fisher-Freeman-Halton exact test: ** *p* < 0.01, *** *p* < 0.001.

## Data Availability

The data are not publicly available due to privacy and ethical restrictions.

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
