# Peer review of "Prospective Assessment of Tumour Burden and Bone Disease in Plasma Cell Dyscrasias Using DW-MRI and Exploratory Bone Biomarkers"

_cancers, 2022, doi:10.3390/cancers15010095_

Round 1
Reviewer 1 Report
It is an interesting study evaluating DW-MRI and bone biomarkers in plasma cell dyscrasias
However I have several comments as follows :
1. Patients with relapsed myeloma are not well clinically characterized. Did they relapsed after first line therapy? If they have several lines of treatment, did they benefit anti-BCMA therapy?
2. There is no information in "Materials and methods section" about the blood collection conditions (Serum samples stored at -20°C or -80°C before analysis?) and the type of ELISA kit used to make analysis (commercial ELISA kit? name of corporation?)
3. Did the patients have cytogenetic assessment? If they did, it will be interesting to know the relation of bone remodeling markers with cytogenetics
Author Response
Dear Reviewer,
We sincerely thank you for your time in reviewing our manuscript titled "Prospective Assessment ofTumour Burden and Bone Disease in Plasma Cell Dyscrasias using DW-MRI and Exploratory BoneBiomarkers" that we submitted to Cancers. Please find below point-by-point responses to comments:
- Patients with relapsed myeloma are not well clinically characterized. Did they relapsed after first line therapy? If they have several lines of treatment, did they benefit anti-BCMA therapy?
Many thanks for raising this important point. We went back to the patient records to characterise the number and nature of myeloma relapses. All were indolent relapses (largely asymptomatic biochemical relapses) without aggressive features, and all were first relapses following first line therapy.
The following has been added under Section 3.1:
"Of 12 patients with relapsed myeloma, all patients had indolent first relapses following first-line therapy.
- There is no information in "Materials and methods section" about the blood collection conditions (Serum samples stored at -20°C or -80°C before analysis?) and the type of ELISA kit used to make analysis (commercial ELISA kit? name of corporation?)
Many thanks for seeking clarification on this. I have added details about the blood collection timing, volume and storage conditions. The ELISA kits were commercial (although I have not received a reply from the researcher who ran the tests, to know what the name of the corporation was).
The following has been added under Section 2:
"Peripheral blood was collected as a morning fasted 20mL clotted (serum) sample and stored at -20C prior to analysis."
Additionally, we have added under Section 2.4 that these ELISA kits were "commercial".
- Did the patients have cytogenetic assessment? If they did, it will be interesting to know the relation of bone remodeling markers with cytogenetics
Many thanks for raising this suggestion - I performed this additional analysis. Unfortunately, there was no statistically significant differences in serum biomarker measurements between those with standard-risk or high-risk cytogenetics [defined as >1 of: t(4;14), t(14;16), t(14;20), gain/amp(1q21) or del(17p)].
The following has been added under Section 3.3:
There was no difference in serum biomarker measurements between patients with standard-risk and high-risk cytogenetics [defined as of: t(4;14), t(14;16), t(14;20), gain/amp(1q21) or del(17p)] (18), in any myeloma (new, relapsed or smouldering) [Mann-Whitney, p>0.05].
We have re-submitted the manuscript with the above changes incorporated. Please do let me know if there are any further edits or information you suggest. Thank you very much again for your time in reviewing our manuscript, and we look forward to hearing from you soon.
Yours faithfully,
Dr Gaurav Agarwal
Reviewer 2 Report
the authors have identified some interesting associations but the number of patients are small in each category of new/relapsed MM, SMM and MGUS. No information on anti myeloma treatment, zolendronate/other biphosphonate/denosumab use, or vit.D status is provided.
Author Response
Dear Reviewer,
We sincerely thank you for your time in reviewing our manuscript titled "Prospective Assessment of Tumour Burden and Bone Disease in Plasma Cell Dyscrasias using DW-MRI and Exploratory Bone Biomarkers" that we submitted to Cancers. Please find below point-by-point responses to comments:
The number of patients are small in each category of new/relapsed MM, SMM and MGUS.
Thank you - we accept that the absolute number of patients is small. We have modified the text to more clearly acknowledge that the study was intended as a pilot feasibility study. We have also softened the language around the study conclusions, to rather be observations, which now require validation in larger cohorts.
The following Sections have been modified to address this point (resubmission attached):
Simple summary
Abstract
Final paragraph of introduction
Discussion
Conclusion
No information on anti myeloma treatment, zolendronate/other biphosphonate/denosumab use, or vit.D status is provided.
Many thanks for raising this point. We have added the number of new and relapsed myeloma patients who were on anti-myeloma treatment. We have also added the bone protection status of patients with any myeloma or MGUS (active, previous or never treatment).
The following has been added under Section 3.1:
- At baseline (or after but before follow-up visit), 13/14 new myeloma and 9/12 relapsed myeloma patients had received active treatment.
- As expected, there was a statistically significant difference in the numbers of patients on bone protection agents at baseline between groups [Fisher-Freeman-Halton exact test, p<0.001], with a greater proportion of new or relapsed myeloma patients on active or previous therapy.
We have re-submitted the manuscript with the above changes incorporated. Please do let me know if there are any further edits or information you suggest. Thank you very much again for your time in reviewing our manuscript, and we look forward to hearing from you soon.
Yours faithfully,
Dr Gaurav Agarwal
Reviewer 3 Report
In this paper Agarwal et al prospectively analyzed a series of bone turnover markers (BTM) together with WB DWI-MRI in a series of 55 patients with MGUS (intermediate-high risk), SMM, NDMM and relapsed MM and 12 healthy volunteers in order to investigate their value in evaluating MM tumour burden and bone disease. The rationale of the protocol is clear and of interest. Even if this is a pilot study with a limited number of patients, the many biases of selection make it impossible to draw any solid conclusions. Too many variables are present in this study: starting from selection of patients shown in table 1: 4 out of 10 MGUS patients have DWI MRI vertebral lesions: are they truly MGUS? Were SMM enroled in the study high or low risk? Were NDMM patients transplant eligible or the cohort was represented by non transplant eligible patients, or both? When combined SMM/MM as therapy responders vs non therapy responders, were also SMM patients under therapy? Concerning relapsing patients, are we dealing with aggressive or indolent relapse?
In addition, it is not acceptable that DWI-MRI performed after starting therapy (how long since starting?), should be considered as basal.
Taking together all the above quoted biases make any interpretation of results hard to get
Minor:
Reference section is very confusing: the same reference is reported twice (i.e. n.12 and n.14); some references do not indicate Authors’ names (i.e. n. 4 and n.25)
Author Response
Dear Reviewer,
We sincerely thank you for your time in reviewing our manuscript titled "Prospective Assessment of Tumour Burden and Bone Disease in Plasma Cell Dyscrasias using DW-MRI and Exploratory BoneBiomarkers" that we submitted to Cancers. Please find below point-by-point responses to comments:
Even if this is a pilot study with a limited number of patients, the many biases of selection make it impossible to draw any solid conclusions.
We have modified the text to more clearly acknowledge that the study was intended as a pilot feasibility study. We have also softened the language around the study conclusions, to rather be observations, which now require validation in larger cohorts.
The following Sections have been modified to address this point (resubmission attached):
Simple summary
Abstract
Final paragraph of introduction
Discussion
Conclusion
4 out of 10 MGUS patients have DWI MRI vertebral lesions: are they truly MGUS?
Thank you for seeking clarification on this. The table details the number of patients with "radiological evidence of previous fracture" based on imaging prior to the study; however, these were judged by the treating clinician to be independent from MGUS, and none of these patients had lytic lesions revealed on DW-MRI in this study.
The following phrase has been added under Section 3.1:
Of 14 patients with MGUS, 4 had radiological evidence of a previous fracture (3 vertebral and 1 nonvertebral), although none of these patients later had lytic lesions identified on DW-MRI.
- Were SMM enrolled in the study high or low risk?
- Were NDMM patients transplant eligible or the cohort was represented by non-transplant eligible patients, or both?
- Concerning relapsing patients, are we dealing with aggressive or indolent relapse?
Many thanks for raising these important points. We have added to Section 3.1, in which we describe the participant demographics, to better characterise our cohort.
The following phrase has been added under Section 3.1:
Of 14 patients with new myeloma, 8 were transplant eligible and 6 were transplant ineligible. Of 12 patients with relapsed myeloma, all patients had indolent first relapses following first-line therapy. Of 15 patients with SMM, 8 had low-risk, 6 had intermediate-risk and 1 had high-risk SMM, as defined by the 2020 IMWG criteria (17).
When combined SMM/MM as therapy responders vs non therapy responders, were also SMM patients under therapy?
Thank you for raising this point. We were interested in broadly comparing the % change in serum biomarkers between baseline and follow-up visit, and benchmarking this against change in paraprotein (categorised as discrete IMWG responder and non-responder categories). We have edited the text to clarify that SMM patients were not under therapy, and added the numbers of new and relapsed myeloma patients who were (the total patient numbers are slightly less than the original numbers recruited due to loss of follow-up).
The following has been added in Section 3.3:
Analysis was performed in a pooled cohort of patients with SMM/MM with paired measurements between baseline and follow-up (in which active treatment was received by 10/11 new myeloma, 7/9 relapsed myeloma and 0/12 SMM patients, with the remaining under observation).
In addition, it is not acceptable that DWI-MRI performed after starting therapy (how long since starting?), should be considered as basal.
Thank you - we accept that this is a limitation of the study. We initially acknowledged it under the Discussion section but have also added a few sentences where the results are described. The range of the number of days between starting chemotherapy and having baseline DW-MRI was 6-81 days.
The following has been added to Section 3.4:
We explored whether MY-RADS analysis of DW-MRI could categorise changes in tumour burden. As discussed under 'Methods', due to clinical urgency, most patients had commenced chemotherapy imminently after diagnosis and before baseline DW-MRI. To minimize variability due to this factor, we included only patients who were on chemo-therapy at time of baseline DW-MRI. Of patients with new or relapsed myeloma with lytic lesions identified on DW-MRI, it had been between 6-81 days since initiating chemo-therapy.
Reference section is very confusing: the same reference is reported twice (i.e. n.12 and n.14);some references do not indicate Authors' names (i.e. n. 4 and n.25)
Apologies for these errors. I have rectified the references.
We have re-submitted the manuscript with the above changes incorporated. Please do let me know if there are any further edits or information you suggest. Thank you very much again for your time in reviewing our manuscript, and we look forward to hearing from you soon.
Yours faithfully,
Dr Gaurav Agarwal
Round 2
Reviewer 3 Report
Again, even if this is a pilot study with a limited number of patients, the many biases of selection make it impossible to draw any solid conclusions.